# Transverse Cracking Induced Acoustic Emission in Carbon Fiber-Epoxy Matrix Composite Laminates

**DOI:** 10.3390/ma15010394

**Published:** 2022-01-05

**Authors:** Zeina Hamam, Nathalie Godin, Pascal Reynaud, Claudio Fusco, Nicolas Carrère, Aurélien Doitrand

**Affiliations:** 1Univ Lyon, INSA-Lyon, UCBL, CNRS UMR5510, MATEIS, F-69621 Villeurbanne, France; zeina.hama@insa-lyon.fr (Z.H.); nathalie.godin@insa-lyon.fr (N.G.); pascal.reynaud@insa-lyon.fr (P.R.); claudio.fusco@insa-lyon.fr (C.F.); 2IRDL, CNRS UMR6027, F-29200 Brest, France; nicolas.carrere@ensta-bretagne.fr

**Keywords:** acoustic emission, numerical simulation, carbon fiber/epoxy matrix composites, laminate transverse cracking

## Abstract

Transverse cracking induced acoustic emission in carbon fiber/epoxy matrix composite laminates is studied both experimentally and numerically. The influence of the type of sensor, specimen thickness and ply stacking sequence is investigated. The frequency content corresponding to the same damage mechanism differs significantly depending on the sensor and the stacking sequence. However, the frequency centroid does not wholly depend on the ply thickness except for the inner ply crack and a sensor located close enough to the crack. Outer ply cracking exhibits signals with a low-frequency content, not depending much on the ply thickness, contrary to inner ply cracking, for which the frequency content is higher and more dependent on the ply thickness. Frequency peaks and frequency centroids obtained experimentally are well captured by numerical simulations of the transverse cracking induced acoustic emission for different ply thicknesses.

## 1. Introduction

Acoustic emission (AE) is a non-destructive technique that can be used to study real-time damage growth in a material. It is based on the detection of acoustic signals originating from a corresponding damage mechanism. Indeed, when damage occurs in the material, a part of the released energy is converted to an elastic wave that can be detected by AE sensors placed on the structure. AE technique is widely used to detect and identify damage mechanisms in composite materials. Numerous studies were set up to try to establish a link between damage mechanisms in composites and the related acoustic emission, for instance, using empirical correlations between the signal and the source mechanism [1,2,3,4,5,6,7,8]. Some authors combined AE with complementary approaches such as infrared thermography [9] or digital image correlation [10]. However, few validations of the AE source labelling damage mechanisms are provided.

Establishing a robust correspondence between the AE sources and the damage mechanisms is not straightforward. Indeed, the AE signals acquired by the sensors are actually the result of the initial source signal that propagated within a given medium and that was transformed by a sensor. As a consequence, the AE signals depend not only on the damage source but also on the specimen geometry, the material properties and the type of sensor and acquisition chain [11,12,13,14,15,16,17]. It is thus essential to consider the contribution of all these elements in order to correctly analyze the recorded waveforms.

Despite the effectiveness of the AE method as a non-destructive approach to characterize damage in composite materials, the above-mentioned limitations were highlighted experimentally. For instance, the experimental results by Morscher et al. [18] and Oz et al. [15] suggest that AE signal contents corresponding to matrix cracking in composite laminates actually depend on the position of the cracked ply in the composite. They showed, for instance, that contrary to prior studies, peak frequency data alone was unable to fully characterize damage initiation [18].

A better understanding of the link between the different damage mechanisms and the resulting AE signals may be established by numerical simulation of the damage-induced AE [19,20,21,22,23,24,25,26,27]. Such simulations require several aspects to be considered, including (i) the damage mechanism acting as an AE source, (ii) the acoustic wave propagation in the specimen accounting for its geometry and material properties and (iii) the sensor and the acquisition chain. Numerical simulations of damage-induced AE were, for instance, studied by Sause et al. [19,20,21], considering either idealized damage source modelling [19] or realistic crack geometries determined from µ-Computed Tomography [21]. AE simulations due to fiber cracking and fiber-matrix debonding in single-fibre specimens were also set up by Hamam et al. [25,26]. In the previously cited works, the influence of the sensor and the acquisition chain was considered either based on the analytical reciprocity method [28,29] or from the multi-physics simulation of the sensor [22,23,24,30,31].

The simulation of AE also required accurate modeling of the damage mechanisms in the composite that act as AE sources. The first damage occurring in composite laminate is transverse crack formation. Parvizi et al. [32] showed that the loading at crack initiation depends on the ply thickness and that the crack either instantaneously crosses the whole ply in the case of thick plies or initiates from the specimen surface and propagates up to an arrest length in sufficiently thin plies. Transverse cracking in composite laminates can be assessed numerically using a dedicated model such as the coupled criterion (CC) [33,34,35,36,37] or Cohesive Zone (CZ) models [38,39]. In addition to transverse crack initiation and propagation in the inner ply, acoustic wave propagation and acquisition chain must also be considered in order to study this damage mechanism numerically.

This work focuses on the numerical simulation of the AE due to transverse cracking in laminate composites. The objective of the work is the experimental characterization and numerical simulation of the influence of the ply thickness and stacking sequence on transverse matrix cracking induced acoustic emission. The experimental setup and the numerical models are presented in Section 2. The influence of the ply sequence and thickness on the AE is studied in Section 3 both experimentally and numerically.

## 2. Materials and Methods

### 2.1. Experimental Settings

Acoustic emission due to matrix transverse cracking in laminate composites is first investigated experimentally. Specimens with different thicknesses are tested under tensile loading; two of which are presented in this work. The plies are made of Hexply^®^ 8552 Epoxy resin reinforced with AS4 carbon fiber (60% volume fraction). Specimen A is a laminate composite with nine plies [0_3_/90_3_/0_3_]; the thickness of one ply is 0.3 mm, so that the total thickness is 2.7 mm. Specimen B contains three plies [0/90/0], its total thickness is 0.9 mm. Both specimens are 250 mm in length and 25 mm in width. Monotonic tensile tests are performed at a 0.5 mm/min displacement rate on an MTS machine (MTS Systems Corporation, Eden Prairie, United States) equipped with a 50 kN force cell. An extensometer is used to measure the mean applied strain.

Four AE sensors are fixed on the specimens to detect signals originating from damage mechanisms occurring within the composites. Two types of resonant sensors are used, namely (micro80, Mistras Group, Princeton, NJ, USA) and (picoHF, Mistras Group, Princeton, NJ, USA). Each couple of sensors are 120 mm apart, on both sides of the specimen (Figure 1a). The coupling agent is vacuum grease. The sensitivity of the sensors, determined by the reciprocity method [28,29], are shown in Figure 1b. They are responsive to different frequency ranges. The sensors are connected to acquisition systems allowing to record signals detected during the tests. The AE acquisition settings are summarized in Table 1. Only the data located between the sensors are kept. The acoustic signature recorded on both sensors C1 and C2 is kept for each source.

For each signal, the waveform is pre-processed as detailed in [40], and both temporal (amplitude, energy) and frequency (frequency centroid, peak frequency and partial powers) descriptors are extracted from the signal and its fast Fourier transform (FFT). For the signal time windowing, the thresholds used in the study are floating thresholds defined by the percentage of maximum amplitude: 0.1% for the beginning of the signal and 5% for the end of the signal.

### 2.2. Simulation Settings

#### 2.2.1. Composite Laminate Model

AE signals induced by transverse matrix cracking are also simulated numerically using finite elements (FE) for composite laminates exhibiting a different number of plies and stacking sequences. Three-dimensional FE models of the specimens are set up using Abaqus™ Standard. Both [0_n_/90_n_/0_n_] or [90_n_/0_n_/90_n_] (n = 1 or 3) composite laminates are modeled, one ply thickness is 0.3 mm. Each ply is considered as a linear elastic transverse isotropic material with the elastic constants given in Table 2 [41,42], where 1 represents the fiber direction, and 2–3 represents the transverse isotropy plane in the local frame of the ply. Local material orientation is defined in each ply to account for the fiber direction. Boundary conditions consist of prescribed displacements at one end of the composite. The symmetry of the studied configuration enables modeling only 1/4 ([0_n_/90_n_/0_n_]) or 1/8 ([90_n_/0_n_/90_n_]) of the composite. A dynamic implicit solution is adopted in order to account for, on the one hand, acoustic wave propagation due to transverse cracking and, on the other hand, transverse cracking nucleation dynamics. The mesh size is refined in the vicinity of the transverse cracking location with at least 10 elements in the ply thickness, resulting in models containing approximately 500,000 degrees of freedom.

#### 2.2.2. Transverse Cracking

Matrix transverse cracking in 90 deg. inner or outer ply is simulated (Figure 2). It can be noted that for [90_n_/0_n_/90_n_], only one outer ply crack is considered. The strain at first transverse cracking depends on the ply stacking and thickness (Table 3), which can be predicted using either the CC [37] or equivalently using the CZ model [43,44].

Transverse cracking modeling consists of progressively unbuttoning all the nodes along the crack surface during a given time *t_c_*. First, a quasi-static loading step is adopted in order to apply the displacement corresponding to the critical strain at first transverse cracking determined using the CC [37]. The tensile strength and critical energy release rate are given in Table 2. Then, the dynamic solution is adopted to describe first the transverse cracking and subsequent acoustic wave propagation, the node unbuttoning step being performed in at least 10 sub-iterations. The maximum time step is set to 10^−7^ s in order to consider a frequency range up to 1 MHz.

#### 2.2.3. Acoustic Emission

Acoustic emission simulation requires considering, on the one hand, the acoustic wave propagation and, on the other hand, the acquisition chain including the sensor effect. Experimental investigation of acoustic wave propagation shows that the amplitude of the wave decreases when going away from the AE source, which highlights a damping of the signal. Damping is defined using Rayleigh parameters $\alpha_{R}$ and $\beta_{R}$ related, respectively, to low- and high-frequency damping. These parameters are identified experimentally by emitting a source signal which is then captured (micro80 sensor) at different distances to the emitting sensor. Two different artificial sources are used, namely, a chirp signal which excites all the frequencies up to 1.2 MHz and a pencil lead break, i.e., the failure of a 2H graphite lead. The damping coefficient $\alpha \left(\omega \right)$ is calculated based on signals *S* detected at two positions $x_{1}$ and $x_{2}$ (Equation (1)).
(1)$$\alpha \left(\omega \right)=-\frac{1}{x_{2}-x_{1}}ln\left|\frac{S\left(x_{2},\omega \right)}{S\left(x_{1},\omega \right)}\right|$$

A numerical simulation is also set up, and the Rayleigh parameters are adjusted in order to correctly represent the experimentally measured damping. Figure 3 shows the damping coefficient as a function of frequency obtained experimentally for both sources and numerically for $\alpha_{R}=10,000$ s and $\beta_{R}=6.10^{-9}$ s^−1^ Rayleigh parameters. It can be noted that above 800 kHz, the experimental and numerical damping coefficients slightly differ, which can be explained by the fact that the micro80 sensor is not very sensitive beyond this frequency.

The sensor and acquisition chain influence are considered similar to the method described in [26]. The resonant sensors are modelled using their transfer function and their aperture effect. The coupling effect is not considered in this study. Simulated sensors are placed on the surface at various locations from the epicenter of the source to the end of the sample. A wideband point contact sensor called perfect virtual point-contact sensor is also investigated. In this case, the detected signal corresponds to the out-of-plane velocity calculated on a single node. The acoustic emission signal is calculated on a line perpendicular to the crack, at different distances from it, in order to calculate the descriptor variations as a function of the distance between the sensor and the transverse crack.

## 3. Results

### 3.1. Experimental Results

Transverse cracking acoustic emission is first studied experimentally with the aim to highlight the influence of the ply thickness and the sensor choice on the signals and corresponding descriptors. The AE activities recorded for [0/90/0] and [0_3_/90_3_/0_3_] specimens and micro80 and picoHF sensors are presented in Figure 4 together with the stress–strain curve. For the thicker specimen ([0_3_/90_3_/0_3_]), first transverse cracking and failure occur for smaller imposed strain than for [0/90/0] specimen (Figure 4a). It is consistent with previous experiments [32] and simulations [33,35,36,37] on similar configurations.

For the [0_3_/90_3_/0_3_] specimen, the numbers and positions of localized sources by both the micro80 and picoHF sensors are similar, resulting in similar variations as a function of strain (Figure 4b). This means that both sensors are responsive to the same sources. Figure 4b shows the cumulated number of localized sources for both specimens obtained with the micro80 sensor. Therefore, the overall acoustic activity is similarly captured by both types of sensors. A deeper insight into the AE results obtained with both sensors can be established by studying the waveforms of each signal detected by the sensors. Temporal and frequency descriptors of these waveforms can be calculated and compared for both sensors. Figure 5 shows the frequency centroid and peak frequency of each AE signal as a function of strain for both sensors for the [0/90/0] specimen.

Even if the AE source is the same, both sensors do not capture the same acoustic information depending on their sensitivity. For instance, the frequency centroid and peak frequency are both larger for signals acquired with the picoHF sensor than those acquired with the micro80 sensor. Therefore, the frequency content of signals detected by the picoHF sensor is significantly higher than the ones detected by the micro80 sensor. This result highlights the dependency of the AE results to the type of sensor used, which was already pointed out by Godin et al. [11].

The influence of the specimen thickness can also be evidenced through the analysis of the signal descriptors. Figure 6 shows the signal amplitude and the frequency centroid (micro80 sensor) as a function of strain.

The range of strain over which signals are acquired strongly depends on the ply thickness. For the [0/90/0] specimen, the signal amplitude mainly varies between 45 dB and 70 dB, especially for strain smaller than 0.6%. For larger strain levels, signals with a larger amplitude up to 90–100 dB are recorded. The corresponding frequency centroid varies between approximately 400–500 kHz for strain smaller than 0.6%, signals with either smaller (down to ~250 kHz) or larger (up to ~700 kHz) frequency centroid are acquired. For [0_3_/90_3_/0_3_] specimen, except for some isolated signals recorded for strains smaller than 0.5%, the range of amplitude 45 to 100 dB and the 200–700 kHz range of frequency centroid are covered by the acquired signals.

### 3.2. Comparison between Experimental and Simulation Results

The assignment of AE signals to a specific damage mechanism requires a large number of experiments and a dedicated supervised or unsupervised classification approach to be set up. However, the first damage mechanism occurring in composite laminates at the mesoscale is transverse matrix cracking. Therefore, the first signals detected experimentally can be compared to those obtained by numerical simulations of first transverse cracking. This corresponds to the signals detected below 0.5% of strain; they are characterized by two acoustic signatures recorded by sensor C1 and C2. The AE signals are calculated in the simulation based on the out-of-plane node velocity on the specimen surface every 5 mm from the source. For the simulated data, the transverse crack is located at 75 mm from the specimen end, and the sensor moves from the epicenter of the source to the end of the gauge length. For experimental data, the sensor is located at position C1 or C2 (Figure 1a), and the transverse cracks are distributed over the gauge length. These configurations are not strictly equivalent, but the obtained signal descriptor variations can nevertheless be compared.

Three configurations are considered in order to account for the influence of the sensor type and ply thickness:(i)[03/903/03] composite with micro80 sensor,(ii)[0/90/0] composite with micro80 sensor,(iii)[0/90/0] composite with picoHF sensor.

#### 3.2.1. [0_3_/90_3_/0_3_] Composite with micro80 Sensor

The comparison between the signals recorded experimentally and obtained numerically is based on the frequency content of the acquired signals. Figure 7 shows the signal frequency centroid variation as a function of the distance to the transverse crack obtained numerically and experimentally with the two micro80 sensors for [0_3_/90_3_/0_3_] specimen.

Numerical results show a clear decrease of the frequency centroid with increasing distance to the transverse crack. The frequency centroid decrease is not as clear for experimental data as for the numerical result, partly because of scattering. Nevertheless, a similar order of frequency centroid magnitude is obtained experimentally and numerically. Figure 7b shows the peak frequency variation as a function of the distance to the transverse crack. The frequency peak obtained experimentally is almost constant, whatever the wave propagation distance. This frequency peak, approximately 400 kHz, is also captured in the simulations far enough from the crack. However, signals acquired at less than 60 mm from the source in the simulation have a larger frequency peak around 500 kHz. Actually, the signal FFT obtained numerically exhibits two main peaks at around 400 kHz and 500 kHz, the former becoming preponderant over the latter for large enough propagation distance and vice-versa.

#### 3.2.2. [0/90/0] Composite with micro80 Sensor

Figure 8 shows the signal frequency centroid variation as a function of the distance to the transverse crack obtained numerically and experimentally with the two micro80 sensors for [0/90/0] specimen. Similarly to the [0_3_/90_3_/0_3_] specimen, numerical results show a clear decrease of the frequency centroid with increasing distance to the transverse crack. The frequency centroid decrease is also observed experimentally, and a good agreement between both data is obtained. An excellent agreement between experimental and numerical results is also observed for the frequency peak variation as a function of the distance to the transverse crack. The frequency peak is almost constant, approximately 350 kHz, whatever the wave propagation distance.

#### 3.2.3. [0/90/0] Composite with picoHF Sensor

The influence of the sensor is now evidenced by comparing experimental and numerical results with the picoHF sensor. Figure 9 shows the signal frequency centroid variation as a function of the distance to the transverse crack obtained numerically and experimentally with the two picoHF sensors for [0/90/0] specimen. Similarly to the micro80 sensor, numerical results show a decrease of the frequency centroid with increasing distance to the transverse crack. The frequency centroid obtained numerically is in good agreement with experimental data except close to the source where the frequency centroid is overestimated. The peak frequency variation as a function of the propagation distance is rather well described by the numerical model. It can be noted that considering a different sensor leads to different magnitudes of frequency centroids and peak frequency (around 600 kHz for picoHF and around 350 kHz for the micro80 sensor), which is observed both experimentally and numerically.

Finally, the numerical model is able to correctly reproduce the frequency content obtained in the signals acquired experimentally. It enables highlighting ply thickness influence on the frequency content, as well as the influence of the type of sensor. In the following, the numerical model is used to study more in detail the influence of the ply thickness and the stacking sequence on transverse cracking induced acoustic emission.

## 4. Influence of the Ply Thickness, Stacking Sequence

We now provide a numerical analysis of the transverse cracking induced acoustic emission with varying stacking sequence and ply thickness. In this section, we consider a perfect contact sensor in order to highlight the influence of each parameter on the acquired signal. We study the influence of transverse cracking occurring either in inner [0_n_/90_n_/0_n_] (n = 1 or 3) or outer [90_n_/0_n_/90_n_] (n = 1 or 3) ply. Transverse cracking is simulated based on a progressive crack node separation during a given time. Figure 10 shows the signals recorded by the perfect contact-sensor on the specimen surface at the transverse crack epicenter for [0_n_/90_n_/0_n_] and [90_n_/0_n_/90_n_] specimens (n = 1 or 3). Noticeable differences are evidenced depending on the ply thickness and transverse cracking location. This is due, on the one hand, to the size of the initiated crack and, on the other hand, to the signal attenuation when propagating in the material. Outer ply transverse cracking results in signals with a larger amplitude than inner ply transverse cracking at the crack epicenter. This is evidenced by inner ply cracking induced signals having larger amplitudes for larger plies and outer ply cracking induced signals having larger amplitudes than inner ply cracking signals. Outer ply transverse cracking induces similar signals at the crack epicenter, whatever the ply thickness (Figure 10a), whereas significantly different signals are induced by transverse cracking depending on the inner ply thickness (Figure 10b). This phenomenon may be explained by the difference in the acoustic wave propagation distance in the specimen thickness for inner ply cracking. Therefore, it is expected that the descriptors of the signals near the crack epicenter are close for outer ply cracking and different for inner ply cracking.

The same damage mechanism may thus lead to signals exhibiting different descriptors depending on their location in the composite and on the composite mesostructure. Frequency descriptors (frequency centroid and partial powers) of transverse cracking recorded signals for the different stacking sequences and ply thicknesses are analyzed as a function of the distance between the sensor and the crack in Figure 11.

Near the crack, the dependency of the frequency centroid on the ply thickness is marked for inner ply cracking and much less pronounced for outer ply cracking (Figure 11a). However, far enough from the crack (distance larger than 60 mm), the same frequency centroid is retrieved whatever the ply thickness for a given stacking sequence. For any distances between the sensor and the crack, the frequency centroid magnitude is significantly different. The analysis of the partial power variation as a function of the distance between the crack and the sensor gives more details about the ply thickness and stacking sequence influence (Figure 11b–d). It concludes that the frequency content of signals induced by outer ply transverse cracking does not depend on the ply thickness since similar partial powers are obtained. The conclusion is quite different for inner ply transverse cracking induced signals. Indeed, larger high (1000–1500 kHz, Figure 11b) and low (0–250 kHz, Figure 11d) frequency content is obtained for transverse cracking in smaller inner plies, whereas most of the frequency content corresponding to larger inner ply transverse cracking lies in 500–1000 kHz interval. Therefore, contrary to outer ply cracking, the influence of the ply thickness on inner ply transverse cracking induced signals is significant, especially on the frequency content. When the source is located in the mid-plane of the specimen, the symmetrical modes are excited (Figure 12a,b). The fundamental longitudinal mode is excited mainly around 500 kHz in [0/90/0] specimen and above 1000 kHz for [0_3_/90_3_/0_3_].

In this case, with decreasing thickness, the spectral contributions above 1000 kHz increase significantly. We can observe a homothety of frequency axis (here by a 3-factor corresponding to the thickness ratio between 3 and 9 plies configurations), resulting in a frequency shift of the excited portion of the dispersion curves. This result highlights that different sensors should be used to follow the cracking properly in these specimens. When the crack is located in the outer ply in [90/0/90] and [90_3_/0_3_/90_3_] specimens, we can observe that the excited modes are very different. It results in a frequency shift of the excited portion of the dispersion curves (below 500 kHz) accompanied by a switch of the nature of the excited modes from longitudinal to flexural (Figure 12c,d). The effect of thickness is not visible for this type of cracking in terms of frequency. In this case, the same sensors are suitable for both thicknesses.

## 5. Conclusions

Experimental characterization of transverse cracking induced acoustic emission in composite laminates highlights the dependency of the AE results to the sensor type and the specimen thickness. Amplitude and frequency centroid of the acquired signals follow two regimes for thinner specimens depending on the applied strain level (below or above 0.7% applied strain), whereas the same range of amplitude/frequency centroid is obtained for thicker specimens.

Frequency peak and frequency centroid obtained experimentally are well captured by numerical simulations of transverse cracking for different ply thicknesses.

The use of different sensors leads to different frequency contents (frequency centroid and frequency range); this experimental observation is well described by the numerical model. For instance, the peak frequency lies around 350 kHz using micro80 and 600 kHz using the picoHF sensor for [0/90/0] specimens.

The influence of the ply thickness on acoustic emission signals recorded at the crack epicenter is significant only for inner ply transverse cracking, whereas similar signals are obtained for outer ply cracking. This result has a direct consequence if classification approaches are set up since the same damage mechanism leads to different signal descriptors depending on the ply thickness and location. It thus raises the following question: should inner ply and outer ply transverse cracking should be considered as two different damage mechanisms in a classification approach?

The influence of the ply thickness on AE signals (on the frequency centroid, for instance) decreases as the sensor is far from the crack; however, the influence of the stacking sequence remains, regardless ofthe sensor location.

The frequency content of signals induced by outer ply transverse cracking does not depend on the ply thickness. Contrary to outer ply cracking, the influence of the ply thickness on inner ply transverse cracking induced signals is significant, especially on the frequency content and on the frequency centroid only close to the crack.

## Figures and Tables

**Figure 1 materials-15-00394-f001:**
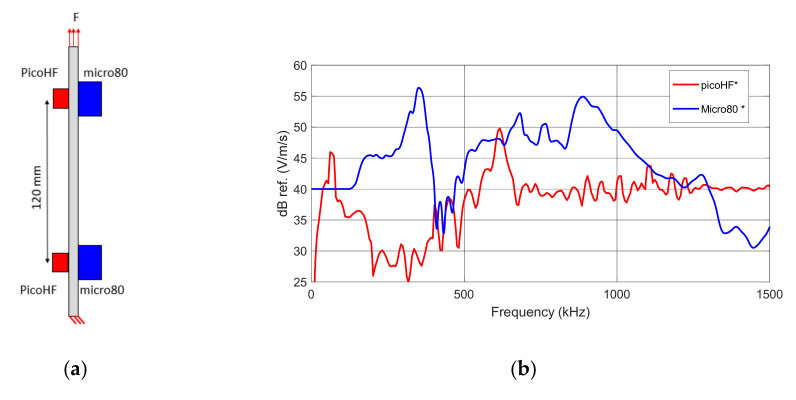
(**a**) Location of the sensors on the specimen (**b**) Frequency sensitivity functions of picoHF and micro80 sensors used to capture acoustic emission during tensile tests in composite laminates (calibration curve obtained on steel block with reciprocity method).

**Figure 2 materials-15-00394-f002:**
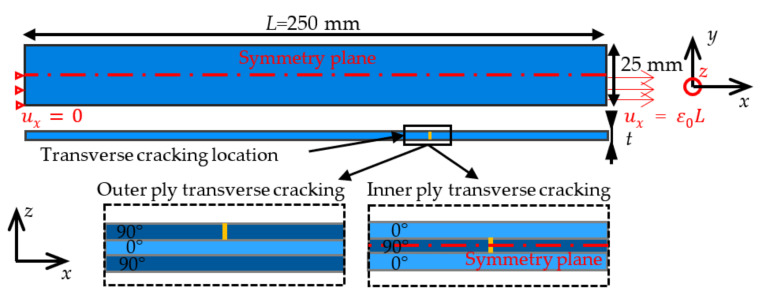
Dimensions of the modeled specimens. Transverse cracking occurs in a 90 deg. Ply with respect to the loading direction.

**Figure 3 materials-15-00394-f003:**
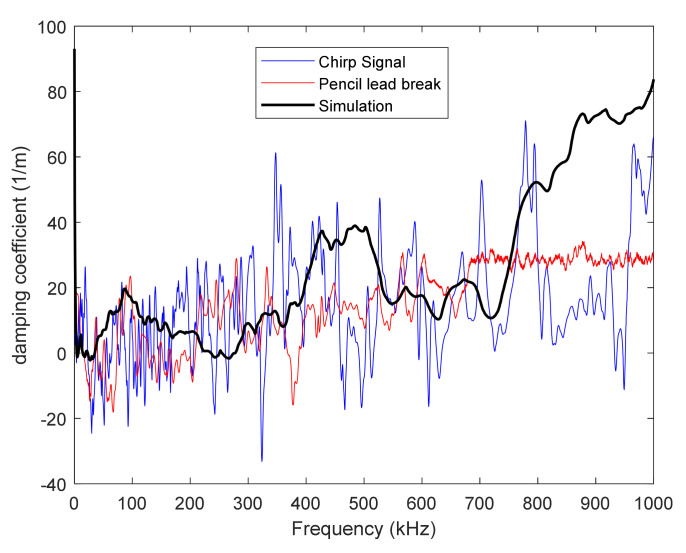
Damping coefficient as a function of the frequency obtained experimentally for (i) emitted chirp and (ii) pencil lead break and (iii) numerically.

**Figure 4 materials-15-00394-f004:**
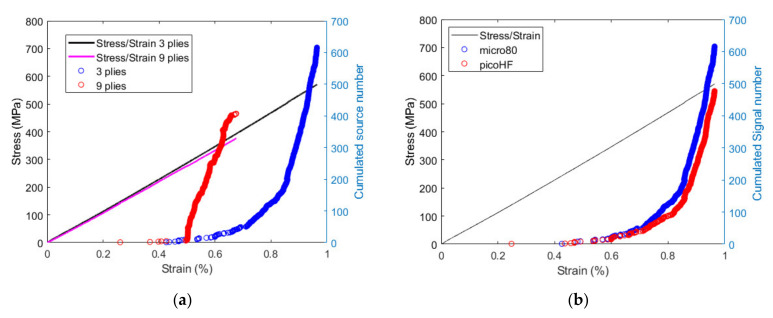
Cumulated localized signal number and stress as a function of strain for (**a**) [0/90/0] and [0_3_/90_3_/0_3_] specimens using the micro80 sensor and (**b**) micro80 and picoHF sensors for [0/90/0] specimens.

**Figure 5 materials-15-00394-f005:**
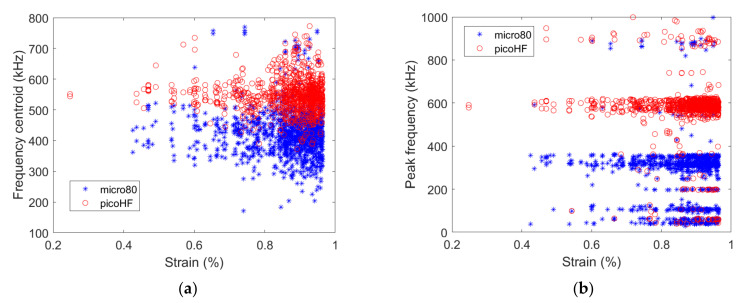
(**a**) Frequency centroid and (**b**) peak frequency as a function of strain obtained for the [0/90/0] specimen (thickness = 0.9 mm) with either micro80 or picoHF sensors.

**Figure 6 materials-15-00394-f006:**
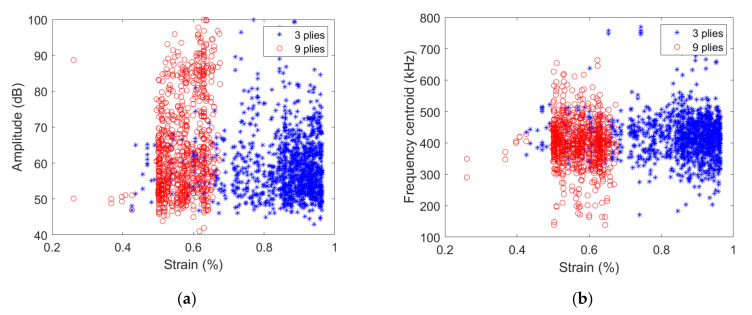
(**a**) Amplitude and (**b**) frequency centroid as a function of strain obtained with micro80 sensor for [0/90/0] or [0_3_/90_3_/0_3_] specimens.

**Figure 7 materials-15-00394-f007:**
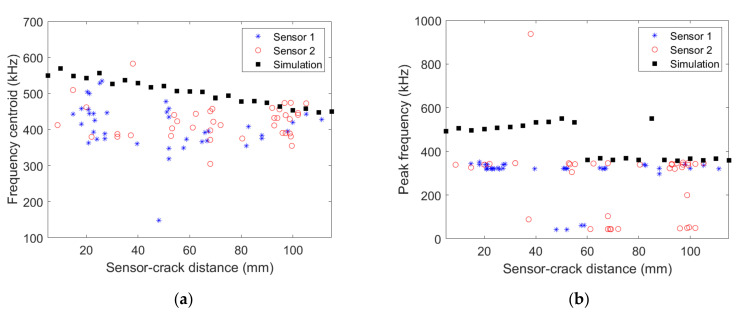
(**a**) Frequency centroid and (**b**) peak frequency as a function of the distance between the AE source and the micro80 sensors obtained experimentally for the [0_3_/90_3_/0_3_] specimen and numerically.

**Figure 8 materials-15-00394-f008:**
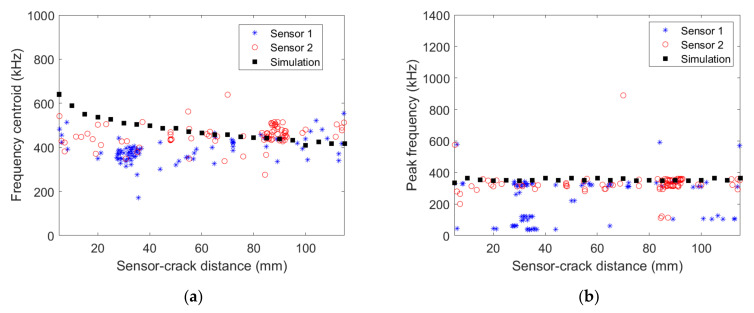
(**a**) Frequency centroid (**b**) peak frequency as a function of the distance between the AE source and the micro80 sensors obtained experimentally for the [0/90/0] specimen and numerically.

**Figure 9 materials-15-00394-f009:**
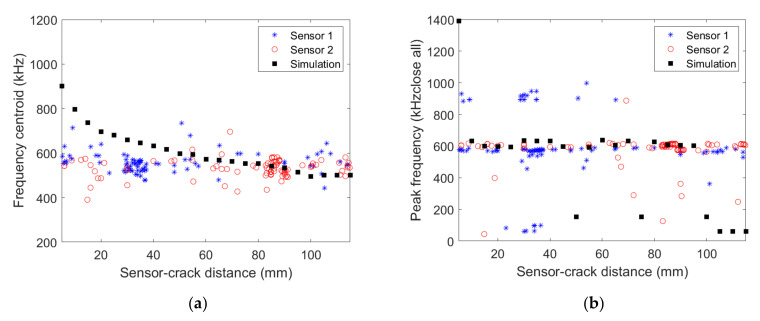
(**a**) Frequency centroid (**b**) peak frequency as a function distance between the AE source and the picoHF sensors obtained experimentally for the [0/90/0] specimen and numerically.

**Figure 10 materials-15-00394-f010:**
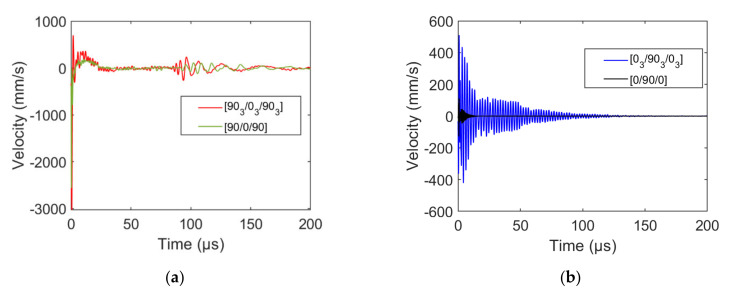
Out-of-plane velocity as a function of time obtained at crack epicenter for (**a**) outer ply transverse cracking in [90_n_/0_n_/90_n_] specimens and (**b**) inner ply transverse cracking in [0_n_/90_n_/0_n_], (n = 1 or 3).

**Figure 11 materials-15-00394-f011:**
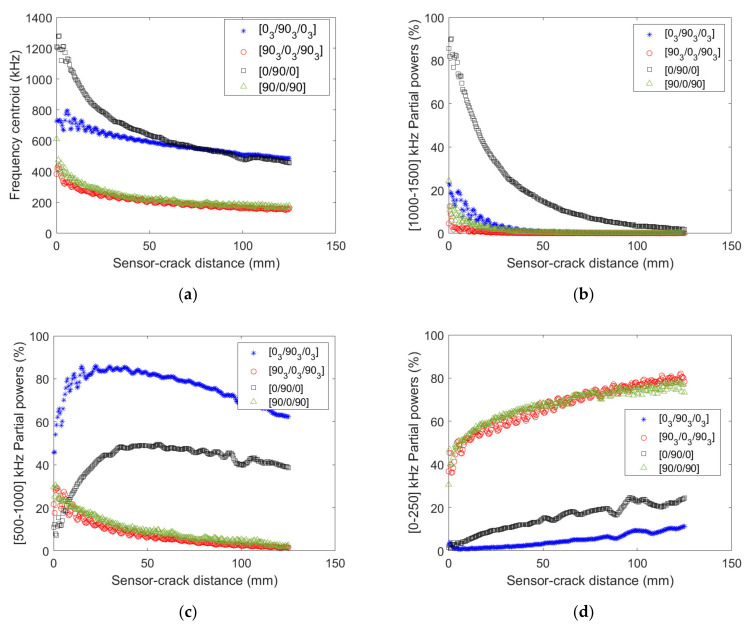
(**a**) Frequency centroid, (**b**) 1000–1500 kHz, (**c**) [500–1000] kHz and (**d**) 0–250 kHz partial powers as a function of distance between the source and the sensor obtained numerically for [0_n_/90 _n_/0 _n_] and [90_n_/0_n_/90_n_] stacking sequences (n = 1 or 3).

**Figure 12 materials-15-00394-f012:**
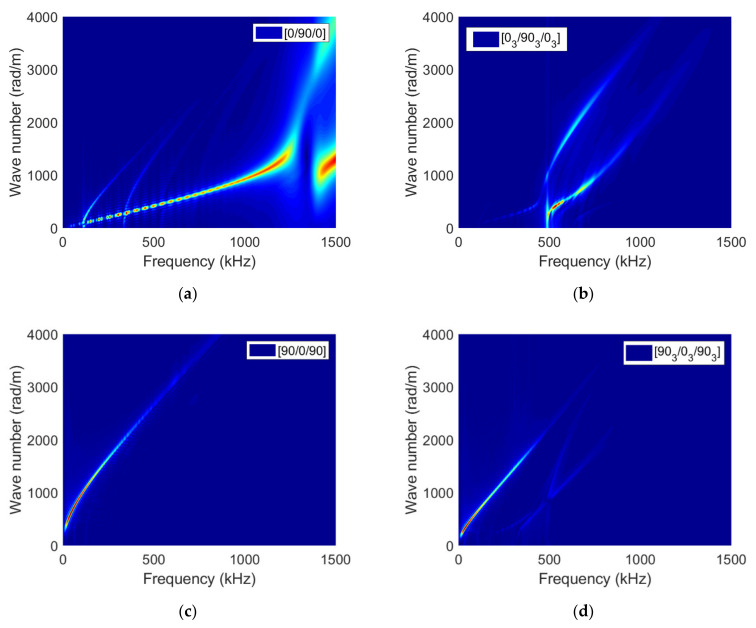
Simulated dispersion curves (wavenumber *k* as a function of frequency *f*) using 2D-FFT of signals recorded along a straight line perpendicular to the crack surface on the specimen top surface for (**a**) [0/90/0], (**b**) [0_3_/90_3_/0_3_], (**c**) [90/0/90] and (**d**) [90_3_/0_3_/90_3_] specimens.

**Table 1 materials-15-00394-t001:** AE acquisition settings.

Sensor	Sensitivity	Gain	Sampling Rate	Threshold	PDT	HDT	HLT	Filter
Micro80	200–900 kHz	40 dB	5 MSPS	38 dB	25 µs	50 µs	1000 µs	20–1200 kHz
PicoHF	500–1850 kHz	40 dB	5 MSPS	38 dB	25 µs	50 µs	1000 µs	20–1200 kHz

PDT: Peak Definition Time; HDT: Hit Definition Time; HLT: Hit Lockout Time.

**Table 2 materials-15-00394-t002:** Elastic and fracture properties of the constitutive plies of the composite laminates.

Properties	Values
$E_{11}\;$(GPa)	127
$E_{22}\;$(GPa)	9.2
$\nu_{12}$	0.302
$\nu_{23}$	0.4
$G_{12}$ (GPa)	4.8
G_c_ (J/m^2^)	248
$\sigma_{c}$ (MPa)	63.9
ρ (kg/m^3^)	1500

**Table 3 materials-15-00394-t003:** Specimen thickness and corresponding imposed strain at first transverse crack initiation.

Configuration	Total Thickness	Strain at First Transverse Crack
[0/90/0]	0.9 mm	0.007
[0_3_/90_3_/0_3_]	2.7 mm	0.004
[90/0/90]	0.9 mm	0.007
[90_3_/0_3_/90_3_]	2.7 mm	0.0025

## Data Availability

The data presented in this study are available on request from the corresponding author.

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
