# Peer review of "Transverse Cracking Induced Acoustic Emission in Carbon Fiber-Epoxy Matrix Composite Laminates"

_materials, 2022, doi:10.3390/ma15010394_

Round 1
Reviewer 1 Report
The objective of the work is to investigate the effects of the ply sequence and the ply thickness on AE signature due to transverse cracking in laminate composites. In this manuscript, both numerical simulations and experimental signal analysis of the AE due to transverse matrix cracking were implemented. Some valuable findings were obtained. However, the current version cannot be published in this journal unless some important corrections could be made .
- This manuscript focus on the numerical simulations of the AE due to transverse matrix cracking. However, there is seldom detailed description of the numerical modelling and the results the progressive damage FEA.
- The conclusion in the abstract was contracted with those described in the section 5 of this manuscript. The authors should check and correct it carefully.
- There are significant difference between the results obtained by the micro80 and picoHF sensors. However, the simulations shows good agreement with the results obtained by these two type sensor. Pleased give detailed explanation of these results or the FEA simulation method.
Author Response
We thank the reviewer for her/his positive appreciation of the work. We made modifications according to the reviewer’s remarks that are highlighted in red in the manuscript.
Remark #1: The FE model was presented in Section 2.2 including details about the geometry and symmetry, the type of solution, time steps, material anisotropic properties and orientations, boundary conditions and mesh description. The acoustic emission simulation is detailed in Section 2.2.3 including wave propagation, signal damping and sensor modeling. It can be noted that transverse cracking is not simulated using a progressive damage model as stated by the reviewer. The AE source modeling is actually obtained by unbuttoning nodes along the crack surface during a given time, as explained in Section 2.2.2.
Remark #2: The reviewer is right concerning the last conclusion given in the abstract, in which we inverted ‘inner’ and ‘outer’ plies We made the appropriate correction in the abstract. Otherwise, the same conclusions are given in the abstract and in Section 5, namely:
(Abstract) “The frequency content corresponding to the same damage mechanism differs significantly depending on the sensor and the stacking sequence.”
(Section 5) “The use of different sensors leads to different frequency contents” and ‘”the same damage mechanism leads to different signal descriptors depending on the ply thickness and location”
(Abstract) “the frequency centroid does not depend on the ply thickness except for inner ply crack and sensor located close enough to the crack.”
(Section 5) “The influence of the ply thickness on acoustic emission signals recorded at the crack epicenter is significative only for inner ply transverse cracking” and “The influence of the ply thickness on AE signals (on the frequency centroid for in-stance) decreases as the sensor is far from the crack”
(Abstract) “Outer ply cracking rather exhibits signals with a low frequency content, not depending much on the ply thickness, contrary to inner ply cracking, for which the frequency content is higher and more dependent on the ply thickness”
(Section 5) “
Remark #3: The significant difference obtained using either micro80 or picoHF sensors are actually obtained both in experiments and in simulations. It can be highlighted by comparing Figures 8 and 9 for instance, which depicts the Frequency Centroid and Peak Frequency obtained using either micro80 or picoHF sensors (experiments and simulations). The results obtained using the two types of sensor are significantly different and in relatively good agreement with experimental results.
Reviewer 2 Report
This paper is sufficiently novel and interesting. The most valuable is conclusions that paper includes characterization of transverse cracking induced acoustic emission in composite laminates in the dependency on sensor type and the specimen thickness.
The origin of this measurement is that frequency peak and frequency centroid obtained experimentally are well captured by numerical simulations of transverse cracking for different ply thicknesses.
This problematic has attracted lots of attention. Among them, the influence of the ply thickness on acoustic emission signals recorded at the crack epicentre is significate only for inner ply transverse cracking, whereas similar signals are obtained for outer ply cracking. Transverse cracking should be considered as two different damage mechanisms in a classification approach. The frequency content of signals induced by outer ply transverse cracking does not depend on the ply thickness.
Paper is well organized, and the investigation is comprehensive and in-depth. Review of topic covered is complete, relevant to paper topic, authors are experienced in this topic and without any gap in knowledge.
Cited references are current (mostly within the last 5 years). The manuscript sounds scientifically. The experimental design is appropriate to test the hypothesis. Based on details the results of paper are reproducible. Figures and tables are interpreted appropriately. Conclusions are consistent with presented arguments.
Author Response
We thank the reviewer for her/his positive appreciation of the work. We are pleased that the reviewer highlighted the main interest and findings of the paper.

Reviewer 3 Report
According to the manuscript with title: “Transverse cracking induced acoustic emission in composite 2 laminates". The submitted work is introducing a new valuable and interesting idea and the given results confirm the idea. This work is suitable for publication in the Journal. I suggest the acceptance after some corrections as follows;
- Add critical results in abstract section
- Give some results with numbers in conclusion
- Add more explanation to experimental work
- Reformulate the novelty of the work in introduction
- The bibliography needs to be improved. Some papers in literature could be taken into consideration such as; Morphological, structural and antibacterial behavior of eco-friendly of ZnO/TiO2 nanocomposite synthesized via Hibiscus rosa-sinensis extract & Laser-assisted for preparation ZnO/CdO thin film prepared by pulsed laser deposition for catalytic degradation & Antibacterial activity of TiO2 doped ZnO composite synthesized via laser ablation route for antimicrobial application & Novel Green Synthesis of Zinc Oxide Nanoparticles Using Orange Waste and Its Thermal and Antibacterial Activity & The influence of calcination temperature on structural and antimicrobial characteristics of zinc oxide nanoparticles synthesized by Sol–Gel method & Green synthesis of high impact zinc oxide nanoparticles.
--It is good to mention and add all these articles that could be important in the introduction section and add to references section;
- Correct typographical errors.
Author Response
We thank the reviewer for her/his positive appreciation of the work. We made modifications according to the reviewer’s remarks that are highlighted in red in the manuscript.
Remark #1: We added in the abstract the following result concerning the comparison between experiments and simulations: “Frequency peak and frequency centroid obtained experimentally are well captured by numerical simulations of transverse cracking induced acoustic emission for different ply thicknesses”.
Remark #2: We quantified the results given in the conclusion with numbers according to the reviewer’s suggestion.
Remark #3: The experimental setup is described in details in Section 2, experimental results and comparison between experiments and simulation are then described in Section 3.
Remark #4: We reformulated in the introduction the novelty of the work which is the experimental characterization and numerical simulation of the influence of the ply thickness and stacking sequence on transverse cracking induced acoustic emission.
Remark #5: All the proposed papers are authored by Menazea and co-workers. After reading the papers, it appears that none of them deal with acoustic emission, neither from experimental or simulation point of view. There is no clear link between these works and the present paper. Therefore, unless the reviewer can provide a sound explanation of how the paper bibliography could be improved by citing these papers, we choose not to consider them.
Remark #6: We corrected typographical errors as suggested by the reviewer.
Reviewer 4 Report
Authors did good work both experimental and numerical models. However, i have a few suggestions to improve the quality of this manuscript as follows:
1) It is good to measure the composite name in the title , keywords as well in abstract as these results are applicable to specific carbon fibre reinforced epoxy composite.
2) In section 2: please use the standard format for the equipments or instruments used in the present study such that (Model, make, city, country).
3) In Table 1: write full form for all abbrevations to make it more reader friendly. Why sensitivity is so large range (200 to 900 kHz)?
Author Response
We thank the reviewer for her/his positive appreciation of the work. We made modifications according to the reviewer’s remarks that are highlighted in red in the manuscript.
1) We added the composite name in the title, abstract and keywords.
2) We now use the standard format according to the reviewer’s comment to describe the different sensors used in this work.
3) We added the meaning of the abbreviations in the table description. Micro80 is a large band sensor, which is responsive in the large frequency range from 200 to 900 kHz.
Round 2
Reviewer 1 Report
This manuscript in this version can be accepted.
Reviewer 3 Report
The authors performed the needed comments and the manuscript is ready now to publish in the journal.